# Bacterial Infections, Trends, and Resistance Patterns in the Time of the COVID-19 Pandemic in Romania—A Systematic Review

**DOI:** 10.3390/antibiotics13121219

**Published:** 2024-12-14

**Authors:** Dan Dumitru Vulcanescu, Iulia Cristina Bagiu, Cecilia Roberta Avram, Licinia Andrada Oprisoni, Sonia Tanasescu, Teodora Sorescu, Razvan Susan, Monica Susan, Virgiuliu Bogdan Sorop, Mircea Mihai Diaconu, Tiberiu Liviu Dragomir, Octavia Oana Harich, Razvan Mihai Horhat, Stefania Dinu, Florin George Horhat

**Affiliations:** 1Department of Microbiology, “Victor Babes” University of Medicine and Pharmacy, Eftimie Murgu Square No. 2, 300041 Timisoara, Romania; dan.vulcanescu@umft.ro (D.D.V.); bagiu.iulia@umft.ro (I.C.B.); horhat.florin@umft.ro (F.G.H.); 2Multidisciplinary Research Center on Antimicrobial Resistance (MULTI-REZ), Microbiology Department, “Victor Babes” University of Medicine and Pharmacy, Eftimie Murgu Square No. 2, 300041 Timisoara, Romania; 3Doctoral School, “Victor Babes” University of Medicine and Pharmacy, Eftimie Murgu Square No. 2, 300041 Timisoara, Romania; 4Department of Residential Training and Post-University Courses, “Vasile Goldis” Western University, 310414 Arad, Romania; 5Department of Pediatrics, “Victor Babes” University of Medicine and Pharmacy, Eftimie Murgu Square No. 2, 300041 Timisoara, Romania; oprisoni.licinia@umft.ro (L.A.O.); tanasescu.sonia@umft.ro (S.T.); 6Department of Internal Medicine II: Diabetes, Nutrition, Metabolic Diseases, and Systemic Rheumatology, “Victor Babes” University of Medicine and Pharmacy, Eftimie Murgu Square No. 2, 300041 Timisoara, Romania; sorescu.teodora@umft.ro; 7Department of Diabetes, Nutrition and Metabolic Diseases, “Pius Brînzeu” Emergency Clinical County University Hospital, 300723 Timisoara, Romania; 8Department of Family Medicine, Centre for Preventive Medicine, “Victor Babes” University of Medicine and Pharmacy, Eftimie Murgu Square No. 2, 300041 Timisoara, Romania; razvansusan@umft.ro; 9Department of Internal Medicine I, Centre for Preventive Medicine, “Victor Babes” University of Medicine and Pharmacy, Eftimie Murgu Square No. 2, 300041 Timisoara, Romania; susan.monica@umft.ro; 10Department of Obstetrics and Gynecology, “Victor Babes” University of Medicine and Pharmacy, Eftimie Murgu Square No. 2, 300041 Timisoara, Romania; bogdan.sorop@gmail.com (V.B.S.); diaconu.mircea@umft.ro (M.M.D.); 11Medical Semiology II Discipline, Internal Medicine Department, “Victor Babes” University of Medicine and Pharmacy, Eftimie Murgu Square No. 2, 300041 Timisoara, Romania; dragomir.tiberiu@umft.ro; 12Department of Functional Sciences, “Victor Babes” University of Medicine and Pharmacy, Eftimie Murgu Square No. 2, 300041 Timisoara, Romania; harich.octavia@umft.ro; 13Department of Restorative Dentistry, Faculty of Dentistry, Digital and Advanced Technique for Endodontic, Restorative and Prosthetic Treatment Research Center (TADERP), “Victor Babes” University of Medicine and Pharmacy, Revolutiei Bv. No. 9, 300041 Timișoara, Romania; horhat.razvan@umft.ro; 14Department of Pedodontics, Faculty of Dental Medicine, “Victor Babes” University of Medicine and Pharmacy Timisoara, Revolutiei Bv. No. 9, 300041 Timisoara, Romania; dinu.stefania@umft.ro; 15Pediatric Dentistry Research Center, Faculty of Dental Medicine, “Victor Babes” University of Medicine and Pharmacy Timisoara, Revolutiei Bv. No. 9, 300041 Timisoara, Romania

**Keywords:** COVID-19, pandemic, bacterial, coinfection, superinfection, trends, resistance, phenotype, Romania

## Abstract

**Background:** The COVID-19 pandemic has intensified concerns over bacterial infections and antimicrobial resistance, particularly in Romania. This systematic review explores bacterial infection patterns and resistance during the pandemic to address critical gaps in knowledge. **Methods:** A systematic review, following PRISMA guidelines, was conducted using databases such as PubMed and Scopus, focusing on studies of bacterial infections from 2020 to 2022. Articles on bacterial infections in Romanian patients during the pandemic were analyzed for demographic data, bacterial trends, and resistance profiles. **Results:** A total of 87 studies were included, detailing over 20,000 cases of bacterial infections. The review found that Gram-negative bacteria, particularly *Escherichia coli* and *Klebsiella pneumoniae*, were the most frequently identified pathogens, alongside Gram-positive *Staphylococcus aureus* and *Enterococcus* spp. Multidrug resistance (MDR) was noted in 24% of the reported strains, with common resistance to carbapenems and cephalosporins. **Conclusions:** The pandemic has amplified the complexity of managing bacterial infections, particularly in critically ill patients. The rise in MDR bacteria underscores the need for stringent antimicrobial stewardship and infection control measures. Continuous monitoring of bacterial trends and resistance profiles will be essential to improve treatment strategies in post-pandemic healthcare settings.

## 1. Introduction

The impact of the COVID-19 pandemic has been researched numerous times, considering that there are more than 430,000 articles in the PubMed database. As the severe acute respiratory syndrome coronavirus 2 (SARS-CoV-2) emerged in late 2019 in China, the situation rapidly evolved into a worldwide pandemic, with the official inception date being in March 2020, as declared by the World Health Organization (WHO) [1].

COVID-19, clinically, was highly contagious, as it could spread even in the pre-symptomatic phase. Also, it was observed that patients with comorbidities, especially elders, were at an increased rate of morbidity and mortality. In the beginning, children were not affected; however, later variants would also be transmitted to them [2].

The pandemic had a profound impact on all healthcare systems worldwide, including Romania, a developing country in Eastern Europe. The country has experienced several waves of COVID-19 infections, with fluctuating numbers in regard to cases, hospitalization, vaccination, and deaths. The virus has disproportionately affected vulnerable groups, notably men, the elderly, and those with underlying health conditions such as hypertension, diabetes, obesity, chronic kidney disease, and cardiovascular diseases [3].

Also worth mentioning are the effects on the psyches of Romanian patients, with increased rates of anxiety, depression, and sense of isolation [4,5,6]. Vulnerable groups were disproportionately affected: elderly individuals faced heightened risk due to comorbidities, children experienced psychological strain, and healthcare workers endured intense frontline pressure [7,8,9,10]. Despite vaccination being a critical tool to curb the pandemic, Romania faced high vaccine hesitancy [8,9,11].

During the pandemic, many facets of the COVID-19 infection were addressed, including biomolecular assays, treatment testing, epidemiological surveying, vaccine safety and efficacy, and the impact and use of laboratory markers such as the C-reactive protein (CRP) [2,3,11,12,13,14,15]. While much attention has been focused on the virus itself, the pandemic also indirectly influenced other critical health concerns, including bacterial infections and antimicrobial resistance. As such, trends of bacterial and antimicrobial resistance in Romania during the pandemic, however, were not that intensively studied.

Amid this public health crisis, bacterial coinfections and secondary infections emerged as significant concerns, further burdening healthcare resources. The presence of such infections often exacerbates the severity of COVID-19 cases, contributing to higher morbidity and mortality rates, especially in hospitalized patients. Part of previous research has mentioned the risk of increased severity due to co-, super-, and nosocomial infections [14]. This issue was amplified during the pandemic due to increased antibiotic use, often empirically prescribed for suspected bacterial coinfections in COVID-19 patients [1,14,16]. Consequently, understanding how the pandemic has influenced bacterial infection trends and resistance patterns is crucial for informing future antimicrobial stewardship policies.

In Romania, the antibiotic resistance situation was already concerning prior to the pandemic. The country reported high resistance rates in common bacterial pathogens, including *Staphylococcus aureus*, *Escherichia coli*, and *Klebsiella pneumoniae*, particularly in hospital settings. The limited implementation of antimicrobial stewardship programs, combined with the over-the-counter availability of antibiotics, contributed to this scenario [17,18].

As such, the aim of this paper is to respond to the call to fill in the gap in regard to the bacterial and antimicrobial resistance trends during the COVID-19 pandemic and to report the overall situation in the country of Romania, as part of the general anti-infectious effort. In order to achieve these goals, the review sought to find articles that contained data on demographics, bacterial infection, and antibiotic use within the territory of Romania.

## 2. Materials and Methods

### 2.1. Study Design

This systematic review follows the Preferred Reporting Items for Systematic Reviews and Meta-Analyses (PRISMA) guidelines. The study has also been registered to the PROSPERO registry, under ID CRD42024546761.

For the selection of the studies, the following databases were searched: PubMed, Scopus, the Cochrane Library, and MEDLINE. The following keyword combinations were used: “COVID-19” or “SARS-CoV-2” or “pandemic”, “bacterial” or “microbial” or “antibiotic” or “antimicrobial”, “infection” or “coinfection” or “superinfection” or “resistance”, “trends”, “Romania”. Secondarily, to check for any missing articles, another search was conducted with the following combinations: “bacterial” or “microbial” or “antibiotic” or “antimicrobial”, “infection” or “trends” or “resistance”, “2020” or “2021” or “2022”, “Romania”. Both search strategies included the filtering out of other review articles, using the various applications of the “NOT” function available on each database, followed by the “review” keyword.

### 2.2. Selection Criteria

Regarding the selection criteria, the following were considered for inclusion: (1) national or international articles that featured Romanian patients infected with a bacterial disease; (2) articles that included the period of interest (POI), specifically, the years 2020, 2021, and 2022, even partially. Exclusion criteria were (1) articles in other language than Romanian or English; (2) articles with incomplete basic data on the patients (e.g., failing to mention dates or links to the COVID-19 pandemic); (3) international articles that did not address the Romanian patients separately/that pooled data along with other countries; and (4) other reviews, meta-analyses, editorials, letters, or commentaries.

Articles were saved using Zotero (v. 6.0.36) software, which was also used for the detection and elimination of duplicates.

### 2.3. Data Extraction

Data extraction was performed by 2 researchers (RS and MS). All titles and abstracts were independently screened by the 2 researchers. When there were any discrepancies, the 2 would reassess the articles together. If doubt still persisted, a senior researcher (FGH) was involved. If there was still uncertainty, the article would be added to the full read list and reassessed afterward.

The following data were extracted: demographic data (number of patients, sex, location of origin, age), clinical features (location of infection, hospitalization days, admittance to the ICU, death), bacteriological findings (type of infection, detection of bacteria and method, antimicrobial testing and method, resistance phenotype), and impact on/of the COVID-19 pandemic. The region was also registered based on the NUTS II document [19].

Data were extracted from article texts, figures, tables, and Appendix A. All data were then saved in an Excel database.

### 2.4. Quality Assessment

Quality assessment and bias control were done independently by 2 researchers (DDV and TS). Similarly, to the data extraction process, if doubt persisted, the senior researcher (FGH) was involved. The Study Quality Assessment Tools published by the NHLBI (available at https://www.nhlbi.nih.gov/health-topics/study-quality-assessment-tools- accessed on 13 June 2024) were employed, as they are specific for study design and have the ability to identify potential flaws in methodology and implementation.

For original studies and controlled trials, this led to a 14-question point-based tool, while for case presentations/series, a 9-question point-based tool was developed. For all studies, 1 positive answer was worth 1 point, while other answers were worth 0 points. Afterward, the obtained total was divided by the maximum possible number of questions. As such, all studies that had a percentage of less than or equal to 33.33% were considered to be of low quality, those between 33.33% and 66.67% (inclusive) were considered to be of fair quality, and those up to 100% were considered to be of high quality.

## 3. Results

### 3.1. Overview

During the search phase, two successive search strategies were implemented, the first one addressing the pandemic period by name (“COVID-19” or “SARS-CoV-2” or “pandemic”). In order to overcome the possibility of missing some articles, that may have not mentioned the pandemic, but focused on patients in the studied timeframe, the second strategy was implemented to address the period by years (“2020” or “2021” or “2022”). The search strategy and its results can be observed in Figure 1, which follows the PRISMA flow diagram.

A total of 2740 articles were obtained, of which 1574 were removed as they were entirely out of the timeframe. From the remaining 1166 articles, 589 were duplicates and were removed. A total of 577 articles were then further screened, of which 425 were removed due to varying reasons. The remaining 152 articles were part of the full read set, out of which 87 final studies were selected.

The results on the types of the articles [20,21,22,23,24,25,26,27,28,29,30,31,32,33,34,35,36,37,38,39,40,41,42,43,44,45,46,47,48,49,50,51,52,53,54,55,56,57,58,59,60,61,62,63,64,65,66,67,68,69,70,71,72,73,74,75,76,77,78,79,80,81,82,83,84,85,86,87,88,89,90,91,92,93,94,95,96,97,98,99,100,101,102,103,104,105,106] and the demographics described by them can be found in the Appendix A. The dataset included 79 original articles (90.80%), seven case reports (8.05%), and one case series (1.15%). The majority of the articles were assessed as being of “fair” quality (n = 61, 70.11%), with 22 (25.29%) rated as “high” quality and 4 (4.60%) as “low” quality. Most studies reported only the number of patients (n = 70, 80.46%), while 10 studies (11.49%) focused solely on the number of samples or episodes, and 8 studies (8.05%) provided data on both. Notably, 20 studies (22.99%) included patients who did not have a confirmed bacterial infection. Across the 70 studies reporting patient counts, a total of 21,527 patients were documented, of whom 14288 (66.37%) had confirmed bacterial infections. In contrast, the 10 studies focusing on samples or episodes reported 18419 cases, while the 7 studies covering both metrics included 3941 confirmed bacterial cases and 6371 samples or episodes.

The study period focused on the pandemic years of 2020, 2021, and 2022, encompassing 38 documents (43.67%). Studies that included at least part of this period were categorized as pre-pandemic (n = 41, 47.13%), post-pandemic (n = 7, 8.05%), or spanning both pre- and post-pandemic periods (n = 1, 1.15%). A total of 34 articles (39.08%) did not explicitly report the number of patients or samples/episodes during the POI. From 50 studies (57.47%), it was estimated that 7503 patients were documented during 2020–2022, while 11 studies (12.64%) reported data on 8757 samples or episodes for the same timeframe.

Regarding age, 48 (55.17%) studies reported data as the mean, with 40 (45.98%) also providing the standard deviation (SD). Based on mean and SD information, the pooled age was 57.94 ± 13.61. There were six case reports that published the ages of the patients. For these patients, the mean was 47.83 ± 20.68. There were 10 (11.49%) documents that described age by median and interquartile range (IQR, n = 6, 6.90%) or range (n = 4, 4.60%). Based on the median and IQR, the pooled age was 52.89 ± 17.39. Lastly, there were 10 (11.49%) articles that provided age data by group. Although the data were somewhat heterogenous, the following overarching grouping can be made: <1 year: 450 (11.33%), 1–18 years: 1329 (33.44%), 18–40 years: 392 (9.89%), 40–60 years: 579 (14.60%), >60 years: 1220 (30.74%).

Sex distribution was not described in 14 (16.09%) articles; 4 (4.60%) articles contained only male subjects and 8 (9.20%) contained only female patients. On the other hand, most studies (n = 61, 70.12%) described patients of both sexes, from which it could be observed that the sex distribution was similar (M: 51.31%, F: 48.69%). However, when accounting for the studies that contained only one specific sex as well, there was a slight shift toward the female sex (M: 40.94%, F: 59.06%).

The determination of the locations of the articles was based on the NUTS II regional areas, as previously mentioned. The regions and their respective studies are as follows: North-West (NW)—1 (1.15%), North-East (NE)—15 (17.24%), West (W)—21 (24.14%), Center (C)—12 (13.79%), south-east (SE)—9 (10.34%), South-West (SW)—7 (8.04%), south (S)—0 (0.00%), and Bucharest (B)—20 (22.99%). Among these, there were three multicentric studies in the same region and two across several regions, resulting in a total of five (5.75%) multicentric articles.

Outcomes of the studies can be found in the Appendix A, and information about the inclusion of the length of hospitalization (n = 40, 45.98% studies), ICU admission (n = 31, 35.63% studies), ICU duration (n = 13, 14.94% studies), and death (n = 35, 40.23% studies). Regarding hospital stays, 26 (65.00%) of the studies reported the mean and SD, resulting in a pooled value of 17.97 ± 9.52 days, while 13 (32.50%) reported median and IQR values, resulting in a pooled value of 11.99 ± 15.76. There was one (2.50%) study that provided its hospitalization value as 14.35 cultures/1000 patient-days.

A total of 10,385 cases (48.24%) involved ICU admissions, with 5666 cases (54.56%) occurring during 2020–2022, as reported in 17 studies (19.54%). Of these, 1146 cases (11.04%) were attributed to confirmed bacterial infections, according to nine studies (10.34%). ICU stay durations were reported in six studies (46.15%) as means with SDs, yielding a pooled value of 8.05 ± 4.49 days, while seven studies (53.85%) provided medians with IQRs, resulting in a pooled value of 11.06 ± 16.90 days. Among 32 studies (36.78%), 2920 patients (13.56%) experienced fatal outcomes, with 651 deaths (22.30%) occurring during the POI, according to 17 studies (19.54%). Deaths from confirmed bacterial infections totaled 150 (5.14%), based on data from five studies (5.75%).

### 3.2. Bacterial Identification

Appendix A details the results regarding bacterial strains and their origins. A total of 81 (93.10%) studies reported the origins of samples/patients for the total declared periods, whereas when we filtered for the POI, only 49 (56.32%) studies declared the locations of origins.

Most studies (total n = 21, 24.14%, POI n = 15, 17.24%) declared their lots as being from multiple wards or from the whole hospital, resulting in 3973 patients and 11,108 samples with identified bacterial origins. The studies described several wards, with infectious disease, pneumology, and the ICU being the most frequent, for both the total cumulated timeframe and the POI, as is observable in Figure 2.

Based on the type of biological sample, data were also heterogenous, with only two studies (2.30%) not reporting this information. Thus, this study collected and compiled data based on samples rather than patients. Most samples across the studies were from urinary cultures (total n = 25, 28.74%, POI n = 15, 17.24%), followed by blood cultures, including hemocultures and central venous catheters (total n = 24, 27.59%, POI n = 20, 22.99%). Other frequently reported sources included wounds and purulent collections (total n= 18, 20.69%, POI n = 16, 18.39%) and stool samples (total n = 17, 19.54%, POI n = 13, 14.94%). Respiratory tract samples were significant, with lower respiratory products like sputum and broncho-alveolar lavage being frequently reported (total n = 16, 18.40%, POI n = 13, 14.94%). Figure 3 highlights the distribution of sample types for the total and pandemic periods.

In the context of the methodology used for bacterial detection, 17 (19.54%) articles did not give details, while the rest mentioned the following: classic culture (n = 29, 33.33%), Vitek 2 systems (n = 23, 26.44%), Matrix-Assisted Laser Desorption/Ionization Time-of-Flight (n = 10, 11.49%), microscopy/histopathological aspects (n = 5, 5.75%), Polymerase Chain Reaction (n = 12, 13.79%), biochemical assays and API galleries (n = 9, 10.34%), immunological and serological assays (n = 16, 18.39%), automated hemoculture systems (n = 4, 4.60%), and other automated identification systems (n = 7, 8.05%).

All studies were selected to have bacterial samples provided; as such, all 87 articles contained some form of data on bacterial etiology, while only 49 (56.32%) provided whether the bacterial strains were identified during the POI. As can be seen from Table 1, across all provided samples, there were 39,823 confirmed bacterial strains, while, based on the 49 manuscripts, there were 12,060 (30.68%) clearly described strains during the POI. Data are presented as totals extracted from the individual studies. There were 10,368 Gram-positive cocci, with 4046 (39.02%) being reported in the 49 manuscripts that properly stratified time periods. Notably, *S. aureus* and CoNS accounted for a significant portion of *Staphylococcus* spp., with 39.50% and 58.48% of cases occurring during the pandemic period, respectively. Among Gram-negative bacteria, *E. coli* and *Klebsiella* spp. were dominant, contributing to 20.59% and 32.2% of pandemic cases. Anaerobes like *Clostridium* spp. (41.52%) and opportunistic pathogens such as *Pseudomonas* spp. (23.57%) and *Acinetobacter* spp. (33.67%) also had notable pandemic contributions.

### 3.3. Antibiotic Resistance (Appendix A)

Regarding antibiotic susceptibility testing, 42 (48.28%) studies declared their methodology, with 22 (25.29%) reporting this information strictly within the POI. The use of automated systems included the Vitek 2 Compact, Microscan, Micronaut, Sensititre, and Bd Phoenix. These data are observable in Figure 4.

Regarding AST protocols for reporting MIC, 37 (42.53%) explained the favored option. A total of 18 (46.65%) studies followed the CLSI standard, 18 (46.65%) followed the EUCAST standard, and 1 (2.70%) reported both. Strictly considering articles pandering to the POI that reported this information (n = 15, 17.24%), six (40.00%) favored the CLSI standard, while the other nine (60.00%) followed the EUCAST protocol. Other information of note is the fact that there were five (5.75%) records that did not explain their AST methods, nor their protocol, yet reported antibiotic resistances.

Information on resistance to antibiotics for all records was declared in 40 (45.98%) articles, with the expanded data being identifiable in Appendix A, while Table 2 presents the summary of antibiotic resistances per drug in each period.

Resistance reports are described further. Penicillins were frequently reported, with ampicillin appearing in 40.00% of studies. Its usage increased significantly post-pandemic, rising from 28.57% pre-pandemic and 30.00% during the pandemic to 66.67% post-pandemic. Amoxicillin was less commonly reported (10.00% overall), but its usage also rose sharply post-pandemic (66.67%), despite being used sparingly during the pandemic (5.00%). Piperacillin, which had limited pre-pandemic usage (4.76%), saw an increase to 50.00% post-pandemic. Combinations like amoxicillin/clavulanate (32.50%) and piperacillin/tazobactam (32.50%) were consistently used, peaking at 33.33% pre- and post-pandemic for amoxicillin/clavulanate, and 40.00% during the pandemic for piperacillin/tazobactam.

Cephalosporins were widely employed, with ceftazidime being the most reported (47.50%). Its usage peaked during the pandemic (65.00%) but declined to 33.33% post-pandemic. Similarly, cefepime (32.50% overall) and ceftriaxone (22.50%) saw their highest usage during the pandemic (40.00% and 25.00%, respectively), with a subsequent decline in the post-pandemic period. Carbapenems, such as imipenem (35.00%) and meropenem (32.50%), followed a similar trend, peaking during the pandemic (55.00% and 45.00%, respectively) but showing reduced usage post-pandemic (33.33% and 16.67%).

Aminoglycosides, particularly gentamicin (47.50%) and amikacin (30.00%), were consistently utilized across all periods. Gentamicin maintained steady usage, reaching 50.00% post-pandemic, while amikacin saw its highest usage during the pandemic (50.00%). Among quinolones, ciprofloxacin was the most frequently reported (52.50%), peaking during the pandemic (55.00%) and remaining widely used post-pandemic (50.00%). Levofloxacin (35.00% overall) followed a similar pattern, with increased use during the pandemic (50.00%).

Other categories showed varied trends. Polymyxins, particularly colistin, were reported in 27.50% of studies, with peak usage during the pandemic (35.00%) and a slight decline post-pandemic (33.33%). Sulfonamides, like cotrimoxazole, were highly prevalent during the pandemic (50.00%) but dropped to 16.67% post-pandemic. Tetracyclines, including tetracycline (32.50%) and tigecycline (17.50%), showed stable usage, with tigecycline usage notably increasing post-pandemic (50.00%). Linezolid and rifampicin saw consistent mentions, while fosfomycin (17.50%) and nitrofurantoin (15.00%) maintained steady usage across periods.

Resistance phenotypes were based on the ones described by Magiorakos et al. [107]. As such, the following resistance profiles were extracted from the studies: multidrug-resistant (MDR), extensively drug-resistant (XDR), pandrug-resistant (PDR), extended-spectrum beta-lactamase producer (ESBL), carbapenem-resistant organism (CRO), methicillin-resistant *S. aureus* and CoNS (MRSA, MRCoNS), and vancomycin-resistant enterococci (VRE). The full list can be observed in the Appendix A. There were 42 (48.28%) studies that reported these data for their whole periods, while 2 (2.30%) explicitly reported no form of multidrug resistance. The list of phenotype resistance by bacteria for all studies for which such data are provided is in Table 3.

Overall, 14 (33.33%) studies reported 1471 (34.55%) unspecified MDR strains, 37 (0.87%) XDR strains, and 42 (0.99%) PDR strains, of which 143 (3.36%) were ESBL strains, and 306 (7.19%) were CRO strains.

*Staphylococcus* spp. was reported in one (2.38%) study with 1 (12.50%) MDR strain, and *Streptococcus* spp., which was featured in two (4.76%) studies, had 56 (11.00%) MDR strains. Regarding *S. aureus*, reported in 21 (50.00%) studies, there were 1657 (33.44%) MDR strains, of which 1655 (33.40%) were MRSA. For CoNS, there were five (11.90%) studies, which observed 300 (38.96%) MDR strains, all being MRCoNS. Regarding *Enterococcus* spp., 16 (38.10%) studies observed 289 (16.64%) MDR strains, with 219 (12.61%) being VRE strains. There were also three (3.70%) high-level aminoglycoside-resistant (HLAR) *Enterococcus* variants, according to one (1.15%) study.

Regarding *E. coli*, 15 (35.71%) studies reported 2269 (35.59%) MDR strains, with 763 (11.97%) strains being ESBLs and 82 (1.29%) being CROs. *Klebsiella* spp. was noted in 17 (40.48%) studies, with 2286 (68.55%) MDR strains, 43 (1.29%) XDR strains, and 1 (0.03%) PDR strain. There were 427 (12.80%) ESBL and 486 (14.57%) CRO strains described. According to only one (1.15%) study, there were 57 (8.07%) Metallo-β-lactamase (MBL)-producing *K. pneumoniae.* For *Proteus* spp., five (26.76%) studies reported 170 (33.53%) MDR strains, 1 (0.20%) XDR strain, and 10 (1.97%) PDR strains. *Salmonella* spp. was featured in two (4.76%) studies, with seven (4.70%) MDR strains, while *Serratia* spp. had one (2.38%) study reporting one (33.33%) MDR strain of ESBL origin. *Enterobacter* spp. featured in two (4.76%) studies with 18 (16.67%) MDR strains, with 3 (2.78%) of them being reported as ESBLs.

*Pseudomonas* spp. was featured in 13 (30.95%) studies, with 683 (46.37%) MDR strains, 9 (0.61%) XDR strains, and 1 (0.07%) PDR strain. Other phenotypes described were 4 (0.27%) ESBL strains and 21 (1.43%) CRO strains. For *Acinetobacter* spp., there were nine (21.43%) studies, which observed 456 (75.12%) MDR strains, 6 (0.99%) XDR strains, and 14 (2.31%) cases of PDR, with 2 (0.33%) being described as ESBLs and 12 (1.98%) as CROs. *R. picketti*, reported in one (2.38%) study, had three (75.00%) MDR strains of ESBL origin.

For *H. influenzae*, one (2.38%) study reported 2 (33.33%) MDR strains, while *H. pylori* had one (2.38%) study reporting 17 (11.49%) MDR strains. For tuberculosis, one (2.38%) study reported two (2.22%) MDR strains. In the “Other” category, featured in four (9.52%) studies, 158 (37.35%) MDR strains and 3 (0.71%) PDR strains were observed.

There were 25 (28.74%) studies that reported these data for the period 2020–2022, while 1 (1.15%) explicitly reported no form of multidrug resistance. The list of phenotype resistance by bacteria for all studies for which such data are provided is in Table 4.

In the unspecified category, seven (28.00%) studies reported 721 (72.98%) MDR strains and 37 (3.74%) XDR strains, of which 48 (4.86%) were ESBL and 77 (7.79%) were CRO strains. For *Streptococcus* spp., one (4.00%) study observed two (1.04%) MDR strains. Regarding *S. aureus*, one (4.00%) study observed 873 (32.08%) MDR strains, of which 871 (32.01%) were MRSA. For CoNS, there were three (12.00%) studies that reported 105 (24.88%) MDR strains, all being MRCoNS. For *Enterococcus* spp., eight (32.00%) studies observed 138 (25.00%) MDR strains, with 124 (22.46%) being VRE.

Regarding *E. coli*, six (24.00%) studies reported 251 (31.85%) MDR strains, with 201 (25.51%) being ESBLs and 1 (0.13%) a CRO. For *Klebsiella* spp., there were 10 (40.00%) studies that observed 707 (62.29%) MDR strains and 37 (3.26%) XDR strains. Also, there were 63 (5.55%) ESBL strains, and 182 (16.04%) CRO strains described. The one (1.15% study describing MBL noted 54 (40.30%) strains for *K. pneumoniae*. *Proteus* spp. was featured in two (8.00%) studies, with 51 (36.43%) MDR strains and 10 (7.14%) PDR strains. *Serratia* spp. appeared in one (4.00%) study, where one (33.33%) MDR strain was reported, being an ESBL.

For *Pseudomonas* spp., there were four (16.00%) studies that reported 70 (32.56%) MDR strains, 8 (3.72%) XDR strains, and 1 (0.47%) PDR strain. Twelve (5.58%) strains were reported as CROs. For *Acinetobacter* spp., there were four (16.00%) studies, which observed 213 (67.19%) MDR strains, 6 (1.89%) XDR strains, and 13 (4.10%) PDR strains. Three (0.95%) strains were reported as CROs.

Regarding tuberculosis, one (4.00%) study reported two (2.22%) MDR strains. Finally, other bacteria appeared in one (4.00%) study, where 85 (39.17%) MDR strains and 3 (1.38%) PDR strains were observed.

## 4. Discussion

### 4.1. Demographics and Outcomes

The demographic data extracted from the studies conducted during the COVID-19 pandemic in Romania reveal a complex picture of bacterial infections that varied significantly across age groups, sexes, and hospital settings. The pandemic period (2020–2022) affected various population segments, with the elderly being disproportionately impacted. Of the total studies reviewed, over 66% of patients with confirmed bacterial infections were older adults, particularly those aged 60 and above, a group that made up 30.74% of the reported cases. This higher prevalence among the elderly likely reflects their increased vulnerability to both COVID-19 and bacterial co- or superinfections [2,108].

In contrast, younger populations (those between 1 and 18 years old) accounted for 33.44% of the total bacterial infections, which represents a significant percentage but is lower compared to the elderly cohort. This is in line with other global reports indicating that, although younger individuals are less likely to suffer severe outcomes from COVID-19, they remain susceptible to bacterial superinfections, particularly in hospital environments [7,109]. Several studies reported data from pediatric and neonatal ICUs, where sex distribution was less explicitly stratified, but the overall trends align with a slightly higher prevalence of infections among female patients [110].

Sex distribution among the patients was fairly balanced, with males representing 51.31% and females 48.69%. However, in studies that exclusively reported on one sex, female patients were more common, comprising 59.06% of cases, suggesting that certain wards, such as obstetrics/gynecology, may have influenced this distribution [108].

The regional distribution of bacterial infections also varied, with the West region (24.14% of cases) and Bucharest (22.99%) being the most heavily represented areas. This uneven distribution could reflect variations in healthcare infrastructure, with better-equipped hospitals in urban centers being more likely to report detailed bacterial data. Additionally, the southern region of Romania had no reported cases. To the best of our knowledge, this is the first review to assess the distribution of bacterial infections during the COVID-19 pandemic by regions.

Outcomes related to hospitalization were severe, with lengthy hospital stays and high levels of ICU admissions being prominent features. On average, patients with bacterial infections spent 17.97 days in the hospital, with variations in ICU admissions and durations [111].

ICU admission was reported in 48.24% of cases, and about 54.56% of these admissions occurred during the pandemic years. ICU stays were lengthy, averaging 11.06 days, indicating the severity of bacterial infections in this cohort, particularly when combined with COVID-19. Mortality data revealed that 22.30% of ICU patients succumbed to bacterial infections, a grim reminder of the fatal potential of nosocomial infections during the pandemic. Among these, confirmed bacterial infections contributed to 5.14% of deaths during the pandemic period, as seen in studies that focused on ICU outcomes. These data are in line with the European literature [111,112].

The analysis reveals that a significant proportion of bacterial infections during the pandemic were reported in ICU settings, with 3687 patients from seven studies in the overall period and 1929 patients from six studies during the pandemic period. This indicates that critically ill COVID-19 patients, particularly those in the ICU, were at high risk for bacterial infections, consistent with other studies that found a high prevalence of bacterial coinfections among ICU patients during the pandemic [113].

The infectious disease and pneumology wards also contributed significantly to the bacterial infection burden, with 2342 patients overall and 1462 during the pandemic period. This reflects the high susceptibility of respiratory systems to bacterial infections, especially in the context of viral respiratory diseases such as COVID-19, which can weaken the immune response and increase vulnerability to secondary bacterial infections [114].

### 4.2. Bacterial Trends

Regarding the types of biological samples, the most commonly tested samples were urinary cultures, accounting for 28.74% of the studies during the overall period and 17.24% during the pandemic period. This is followed by blood cultures and wound and purulent collections, which remained consistent across both periods. These findings align with global trends, where urinary and blood cultures are frequently used to detect bacterial infections in hospitalized patients [115].

A large proportion of the studies also focused on lower respiratory tract samples, which include sputum, tracheal aspirates, and BAL, crucial for diagnosing respiratory-related bacterial infections in COVID-19 patients. Respiratory samples were featured in 18.40% of the studies during the overall period and 14.94% during the pandemic period. This emphasis on respiratory samples is not surprising, given the respiratory complications associated with COVID-19 and the need to diagnose bacterial coinfections in patients with severe respiratory symptoms [116].

In terms of bacterial detection methodologies, this study highlights a diverse range of techniques, from traditional culture methods (33.33% of studies) to more advanced methods such as PCR and MALDI-TOF. The shift towards automated and molecular diagnostic techniques, such as Vitek 2 systems and automated hemoculture systems, reflects the increasing demand for rapid and accurate bacterial identification during the pandemic [115,117].

Regarding bacterial species, the most important infections were due to S. aureus, CoNS, *Enterococcus* spp., Enterobacterales (*E. coli*, *Klebsiella* spp.), *Pseudomonas* spp., *Acinetobacter* spp., and *Clostridium* spp.

Among the *Staphylococcus* spp., *S. aureus* emerged as a significant concern, accounting for 39.50% of strains reported during the POI. The increased infection rates among these vulnerable patients further complicated treatment outcomes, exacerbating the risks of severe complications. This trend reflects the global challenge posed by *S. aureus*, both as a coinfection or as a superinfection in ICU settings, where critically ill patients were generally disproportionately affected [14,113,116,118,119,120,121,122,123,124,125,126,127,128]. Among these studies, of note are studies from Hungary [118], Poland [123], Italy [124] Greece [125], and the Netherlands [127]. Additionally, CoNS, often linked to infections from indwelling medical devices, saw a notable rise, with 58.48% of CoNS infections occurring during the pandemic, further underscoring the vulnerability of patients in ICUs and those with implanted medical devices, with elevated trends being reported during the pandemic by the international literature [113,118,119,121,122,123,124,127].

The *Enterococcus* spp. was also significant, with *E. faecium* and *E. faecalis* accounting for 36.18% and 14.40% of the infections, respectively. Notably, E. galinarum was observed only during the pandemic, although with minimal representation (two strains). These bacteria are important due to their high resistance to vancomycin, which can be problematic, especially in the hospital and ICU settings. The elevating trends are also noted by other researchers [14,113,121,122,125,126,127,128,129,130,131].

On the Gram-negative side, *E. coli* was the most prevalent, accounting for 20.59% of cases identified during the POI. This bacterium is a common cause of urinary tract infections, particularly in wards managing gastrointestinal and respiratory complications, and also has been described in bloodstream infections in patients from the ICU [116,118,119,120,121,122,124,127,131,132,133].

*Klebsiella* spp. demonstrated a substantial presence, with 32.20% of cases recorded during the POI. Studies by Hammoudi et al. [16], Tang et al. [116], Ntziora et al. [121], and Ahmed et al. [134] showed a notable increase in *K. pneumoniae* infections, particularly in ICU settings, where it was a leading cause of VAP and bloodstream infections. This aligns with findings from the European and international literature, which highlighted the pathogen’s role in nosocomial outbreaks exacerbated by pandemic-related healthcare strains [14,113,119,120,122,124,126,127,128,131,133].

Other Enterobacterales of note were *Proteus* spp. (56.05% mentioned in the pandemic), *Serratia* spp. (57.58% mentioned in the pandemic), and *Enterobacter* spp. (mentioned in 18.62% of cases during the POI). The pandemic exacerbated several risk factors for Enterobacterales infections (prolonged hospital stays, increased use of invasive devices such as catheters and ventilators, and high patient load in ICUs), as reported by Hammoudi et al. [16], Velásquez-Garcia et al. [126], as well as the other global literature [113,122,123,128,133,135].

Among non-fermenting NFBs, *Pseudomonas* spp. (23.57%) and *Acinetobacter* spp. (63.95%) were prominent. Internationally, Hammoudi et al. [16] reported a similar trend, with *P. aeruginosa* being a significant cause of VAP in ICUs. This pathogen’s high adaptability and resistance levels contributed to its continuous prevalence. Other international studies described mixed trends in different products [14,113,116,119,120,121,122,124,125,126,127,128,132,133]. Trends in *A. baumannii* also reflect international observations, especially due to its elevated capability for multidrug resistance [14,113,116,119,121,135]. The high prevalence in ventilated patients highlights the urgent need for targeted interventions, according to Hammoudi et al. [16] and Sleziak et al. [133]. Also noteworthy is the mention by Orosz et al. [118] in neighboring Hungary, and Petrakis et al. [125], due to their similar healthcare systems.

In addition to Gram-positive and Gram-negative pathogens, anaerobic bacteria, particularly *Clostridium* spp., played a substantial role, accounting for 40.92% of the total bacterial burden. This significant contribution underscores the importance of considering a wide spectrum of bacterial pathogens in COVID-19 patients, especially given the potential for polymicrobial infections in critically ill individuals [132,136]. On a European level, an increase in bacterial isolates, compared to pre-pandemic years, was observed, especially in *Acinetobacter* spp., *E. coli*, *K. pneumoniae*, *S. aureus,* and *Enterococcus* spp. [131].

### 4.3. Antibiotic Use and Resistance Patterns

The rise in antibiotic use during the pandemic raises concerns about antibiotic resistance, as many antibiotics were used empirically due to the high risk of secondary bacterial infections. This is important to know, as Romania has one of the highest rates of overuse of antibiotics in Europe [127]. Compared to other European nations with more robust and better-funded healthcare systems, Romania experienced poorer outcomes in managing bacterial infections [137]. The healthcare system saw a notable increase in healthcare-associated infections, particularly bloodstream and respiratory infections in ICU settings. These infections were often linked to invasive procedures such as intubation and catheterization, which were critical for managing severe COVID-19 cases [1,2,124,127].

The global reliance on broad-spectrum antibiotics was pivotal in managing severe bacterial coinfections during the COVID-19 pandemic, particularly in ICU settings. This trend, as highlighted in the 33 studies, mirrors findings from Wilczyk-Chrostek et al., who noted the widespread use of empirical therapy to combat ventilator-associated pneumonia [138]. Targeted therapies like vancomycin for MRSA and azithromycin for its anti-inflammatory properties were also crucial, aligning with international practices. However, Palaiopanos et al. caution against the potential exacerbation of antimicrobial resistance due to the extensive use of these agents in acute care hospitals [139]. Also of note is an increasing trend towards resistance to carbapenems and even colistin, as also noticed by Petrakis et al. [125].

The discussion on AST during the COVID-19 pandemic highlights the methodologies used in 42 studies (48.28%). Automated systems such as the Vitek 2 Compact, Microscan, and Sensititre were used in 34.49% of cases. These automated systems were particularly valuable in pandemic settings, offering rapid, standardized results crucial for managing bacterial coinfections in critically ill patients, such as those with VAP or bloodstream infections [16,119,120,121]. Gajic et al. underscored the comprehensive adoption across Europe of automated AST systems, noting their value in maintaining diagnostic precision under pandemic pressures [140]. Similarly, Karagiannidou et al. reported widespread use of these technologies in Greek ICUs [141]. PCR technology for detecting antibiotic resistance phenotypes was mentioned in only three studies (3.45%), reflecting its use in identifying specific genetic markers of resistance, though it was not as widely adopted. This may reflect resource constraints or the preference for faster, broader testing methods during the pandemic [82,122].

Manual methods, including disk diffusion, were also prevalent (in 25.29% of studies). While automated systems offer speed, manual methods remain essential in settings where access to advanced technology is limited. Methods such as E-tests and microdilution were also reported, in four studies each; they provide more precise minimum inhibitory concentration (MIC) data in complex cases like multidrug-resistant infections [130]. AST reporting protocols for MICs were divided equally between the CLSI and EUCAST standards in the beginning, with EUCAST being more commonly used in studies pertaining to the POI. This reflects the current transition in our country from one standard to the second, similar to other European countries [142]. However, five (5.75%) studies failed to declare any specific AST methodology, raising concerns about the reliability of their antibiotic resistance data.

According to 48.28% of the studies that covered several forms of resistance, MDR strains were prevalent, making up 24.00% of the strains reported in 14 studies. Out of the studies focused solely on the 2020–2022 period, 28.74% reported 60.32% MDR strains, 3.74% XDR strains, 4.86%, ESBL strains, and 77 (7.79%) CRO strains. The prevalence of resistant strains in this period mirrors the trends seen in the overall dataset, with *Klebsiella* spp. and *Acinetobacter* spp. continuing to be significant contributors to the burden of antibiotic resistance. *Acinetobacter* spp. alone accounted for 213 (67.19%) MDR strains and 13 (4.10%) PDR strains during the pandemic period, highlighting its ongoing role as a critical nosocomial pathogen. Among the pathogens reviewed, Gram-negatives such as *Klebsiella* spp., *Pseudomonas* spp., and *Acinetobacter* spp. exhibited particularly high MDR rates, reflecting global trends [14,113,124].

*Klebsiella* spp. was present in 17 studies, with 68.55% being MDR strains, 1.29% XDR, and 0.03% PDR. Similar to research by Ahmed et al., this study also noticed an increase in resistance towards aminoglycosides [134]. This species also contributed significantly to ESBL (12.80%) and CRO (14.57%) cases, indicating its pivotal role in carbapenem-resistant Enterobacterales (CRE). Such resistance patterns complicate treatment and increase mortality, especially in critical care settings [16,116,120,124]. Contrary to the majority of international research, the study by Altorf-van der Kuil et al. mentioned a decrease in ESBL and CRO strains in the Netherlands [127].

*Pseudomonas* spp. had 46.37% MDR strains across 13 studies, with XDR and PDR rates at 0.61% and 0.07%, respectively. Despite being a non-fermenting Gram-negative pathogen, its resistance to carbapenems (1.43% CROs) and extended-spectrum beta-lactamase production (0.27%) highlights its adaptability in hospital environments, particularly ICUs, where it frequently causes ventilator-associated pneumonia and bloodstream infections [16,116,124,126,133]. This trend, however, has been decreasing in some areas, such as the Netherlands [127].

*Acinetobacter* spp. demonstrated an alarming 75.12% MDR rate in nine studies, with 0.99% XDR and 2.31% PDR. Despite being a non-fermenting Gram-negative pathogen, its resistance to carbapenems (1.43% CROs) and extended-spectrum beta-lactamase production (0.27%) highlights its adaptability in hospital environments, particularly ICUs, where it frequently causes ventilator-associated pneumonia and bloodstream infections [16,116,121,133].

*E. coli*, a common pathogen in both community and healthcare-associated infections, was reported in 15 studies, with 35.59% being MDR strains. Notably, 11.97% of strains were ESBLs, and 1.29% were CROs. This resistance profile aligns with global concerns, as well [116,120,131], with the exception of the Netherlands [126]. *Proteus* spp. was reported in five studies, with 33.53% being MDR strains, 0.20% XDR, and 1.97% PDR. Despite its lower prevalence, the resistance profile is concerning due to *Proteus* spp.’s role in urinary tract infections [122] and VAP [133], which can escalate to more severe systemic infections if not appropriately managed.

Other Enterobacterales reported in this study included *Salmonella* spp. (4.70% MDR), *Enterobacter* spp. (16.67% MDR, 2.78% ESBLs), and *Serratia* spp. (2.78% MDR and ESBLs). MDR *Salmonella* spp. during the pandemic was previously observed by Abro et al.; their study even mentioned XDR cases in Pakistan [143]. *Enterobacter* was also cited as having been MDR, especially of the CRO type, in France [144]; however, the study of Hafiz et al. showed a decrease in resistance as the pandemic went on, in Saudi Arabia [145]. Lastly, *S. marcescens* was observed as being increasingly ESBL during the pandemic in the Czech Republic [146].

These pathogens underscore the gravity of the AMR crisis across Europe, where countries face common challenges yet observe variability in resistance patterns due to differences in healthcare infrastructure, funding, and antibiotic stewardship programs [11,147]. Romania, along with countries such as Bulgaria and Hungary, faces particular challenges in managing AMR due to socioeconomic factors, healthcare funding constraints, and infrastructure limitations.

Carbapenem-resistant Enterobacterales infections have surged, with ICU infection rates reaching 47.2% in some cases, emphasizing the heightened risk for patients in critical care [128]. In 2020, the resistance rates for *A. baumannii* and *K. pneumoniae* in Romania were among the highest in Europe. Compared to neighboring countries such as Hungary, Bulgaria, and North Macedonia, this resistance was slightly elevated [148].

On a broader level, *Staphylococcus* spp. was reported in one study (2.38%), with 12.50% of the strains being MDR. Across 21 studies, 33.40% of *S. aureus* strains were identified as MRSA. These findings align with some global trends, highlighting MRSA’s association with invasive procedures, such as mechanical ventilation and catheterization, which are common in critically ill patients [14,113,116,120,126,149], while most Western European data show a decline in *S. aureus* infections during the pandemic compared to pre-pandemic years. However, countries with healthcare systems comparable to Romania’s, such as Serbia, Croatia, and Poland continue to also report high rates of MRSA [123,148]. Even some Western European countries, such as Italy [124], Greece [125], and the Netherlands [127] have reported elevated incidences of MRSA.

Not as prevalent as in *S. aureus*, resistance in *Streptococcus* spp. was observed in 4.76% of studies. This bacterium remains of interest as it was frequently observed as a coinfection in COVID-19 patients, especially with *S. pneumoniae*. Yet, generally, the incidence and resistance rates have been observed to have decreased in Europe [148].

In 12.00% of studies that covered the pandemic, a total of 105 MDR-CoNS isolates were identified, with MRCoNS accounting for all of these strains. Al-Nsour et al. [150] and Serra et al. [151] further highlight the role of MRCoNS in the hospital setting, while Kim et al. discuss the increase in teicoplanin-resistant variants of *S. epidermidis* [152]. Slabisz et al. reported similar challenges in the context of the COVID-19 pandemic in Poland [123].

Last but not least, VRE poses a critical challenge in healthcare environments, especially among COVID-19 patients who are already vulnerable to severe infections. In 32.00% of studies, 138 MDR *Enterococcus* spp. strains were identified, with most (89.86%) being VRE. These figures underscore the significant burden of VRE, which limits treatment options and complicates patient management, a trend also emphasized by Hammoudi et al. [16] and Khan et al. [130].

VRE strains were identified in 219 (12.61%) cases of *Enterococcus* spp. VRE poses a significant challenge in healthcare settings due to limited treatment options, particularly in COVID-19 patients already compromised by severe infections. Similar trends were highlighted by Khan et al. [16], Hammoudi et al. [130], and Slabisz et al. [123], as well as other international studies [116,125,129.150]. In 2022, the European region reported a population-weighted mean vancomycin resistance rate of 16.8% in *E. faecium*, with considerable variability across countries, ranging from 0% to 56.6%. Notably, nations such as Lithuania, Bosnia and Herzegovina, Serbia, and Romania reported some of the highest resistance rates, reflecting shared healthcare infrastructure challenges [148].

### 4.4. Limitations

The present study, like any research endeavor, has several limitations that should be considered when interpreting its findings. First, while the databases used were searched extensively, there is a possibility that relevant articles were overlooked. This may be due to variations in indexing practices, incomplete database coverage, or search terms that did not capture all pertinent studies. Furthermore, four potentially relevant studies could not be accessed due to availability issues, which may have impacted the comprehensiveness of the data.

Another notable limitation involves the heterogeneity of data across the included studies. While some studies provided detailed bacterial isolate identification and AST results, others were less comprehensive, omitting data on certain bacterial strains or resistance profiles. Some studies focused on specific bacterial pathogens, while neglecting to report resistance phenotypes for other isolates. This inconsistency in reporting limits the ability to draw broad, generalizable conclusions about resistance trends and bacterial prevalence. Moreover, resistance phenotypes, although critical for assessing antimicrobial resistance, were not consistently evaluated, even in studies that performed AST. This heterogeneity in data quality complicates efforts to synthesize findings across studies and weakens the potential for robust comparisons. Another notable limitation may be the lack of long COVID cases, which were not specifically sought.

Additionally, as this review predominantly focuses on bacterial infections and resistance trends within Romania, the findings may not be entirely representative of other countries or regions. Certain areas within Romania were better represented than others, leading to potential geographic bias in the data. This limits the applicability of the results to regions with different healthcare infrastructure, infection control practices, and bacterial resistance patterns.

Despite the effort to mitigate bias, variability in study quality remains a concern. Two researchers independently assessed the quality and potential bias of the included articles, but subjective judgment and potential misclassification may still affect the reliability of the conclusions. The authors acknowledge the need for further research, particularly in the form of a meta-analysis, which would offer a more systematic and quantitative synthesis of the data. A meta-analysis would improve data accuracy by addressing inconsistencies in study design and reporting, ultimately allowing for more precise estimates of bacterial prevalence and resistance trends.

Finally, the study underscores the importance of a more standardized approach to data collection and reporting. The current variability in data presentation, bacterial identification, and resistance stratification across studies hinders the development of comprehensive analyses and robust, evidence-based interventions. Moving forward, adopting consistent reporting standards would not only enhance the accuracy of research findings but also improve national surveillance efforts, inform treatment guidelines, and strengthen antimicrobial stewardship programs. Achieving these goals would be crucial for better controlling bacterial infections and combating antimicrobial resistance on both national and global levels.

## 5. Conclusions

This systematic review aimed to capture the essence of bacterial and antibiotic trends in Romania, in the time of the COVID-19 pandemic. The comprehensive search strategy employed yielded a robust dataset of 87 studies, with great heterogeneity. The mean age of the patients studied was around 58 years, with nearly equal sex distribution, and a high representation of the West (24.14%) and Bucharest (22.99%) regions. Although some regions had minimal or no representation, a small proportion of studies (5.75%) were multicentric, ensuring a broader regional coverage. In terms of clinical outcomes, a substantial portion of the studies addressed critical metrics such as length of hospitalization, ICU admissions, and mortality. The average hospital stay was almost 18 days, while over half of these ICU admissions occurred during the peak pandemic years, adding up to another 11 days of ICU stay. Of the overall patients, 5.14% faced fatal outcomes due to bacterial infection alone during the pandemic, while 22.30% of such cases were first admitted to the ICU.

Regarding sampling information, the ICU, pneumology, and infectious disease wards were the most prevalent. Urinary cultures were the most frequently collected samples, followed by blood cultures, wounds, and purulent collections. Regarding bacterial identification, classic culture techniques were the most commonly reported, followed by the use of Vitek 2 systems and MALDI-TOF for bacterial identification. Other advanced techniques, such as PCR and immunological assay, were also described. The diverse range of bacterial pathogens identified, including *Staphylococcus aureus*, *Klebsiella pneumoniae*, and *Acinetobacter baumannii*, highlights the spectrum of bacterial threats faced in Romanian hospital environments.

The analysis of antibiotic use during the pandemic reveals a heavy reliance on broad-spectrum antibiotics, with the most commonly used antibiotics being penicillins, cephalosporins, and carbapenems. The emergence of resistance, particularly due to the overuse of empiric antibiotics, raises concerns about the sustainability of current antibiotic strategies in our country.

The trends in antibiotic resistance identified during the pandemic are particularly concerning, with MDR, XDR, and PDR bacterial strains becoming more prevalent, particularly among *Klebsiella* spp., *Pseudomonas* spp., and *Acinetobacter* spp., which also showed alarmingly high levels of resistance (72.82%). The high incidence of MRSA/MRCoNS and VRE further complicates treatment options, particularly in critical care settings, as well as ESBLs and CROs, with *Klebsiella* spp. being a significant contributor to both categories. The data from Romania reflect a broader global challenge, wherein antibiotic resistance is exacerbated by increased antibiotic use during health crises like the COVID-19 pandemic.

Current variations in data description and insufficient stratification of bacterial and resistance trends hinder a comprehensive analysis and the development of effective, evidence-based interventions. As part of the antimicrobial effort in our country, a concerted shift towards a unified and methodologically sound system is essential for advancing our understanding of antibiotic resistance trends and optimizing public health responses in Romania.

## Figures and Tables

**Figure 1 antibiotics-13-01219-f001:**
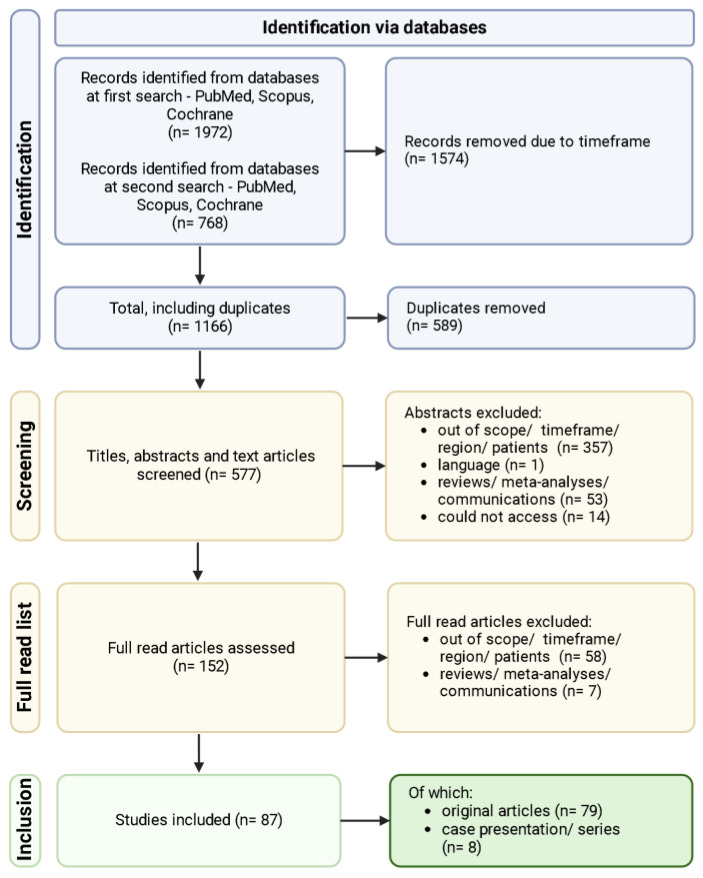
PRISMA flow chart.

**Figure 2 antibiotics-13-01219-f002:**
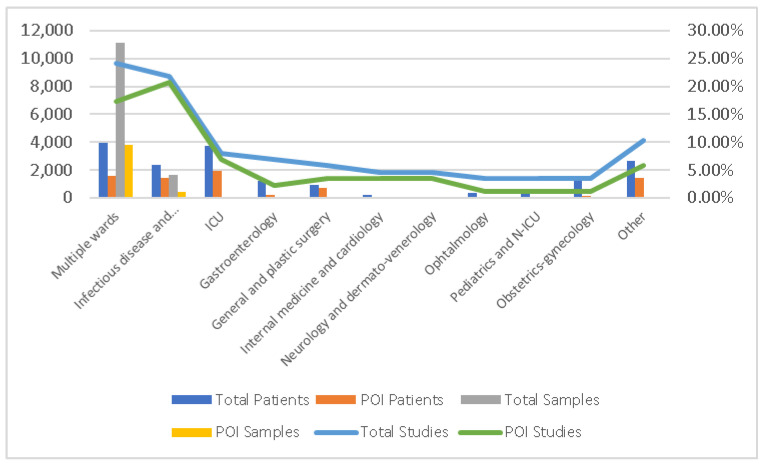
Bacterial isolates in hospital wards.

**Figure 3 antibiotics-13-01219-f003:**
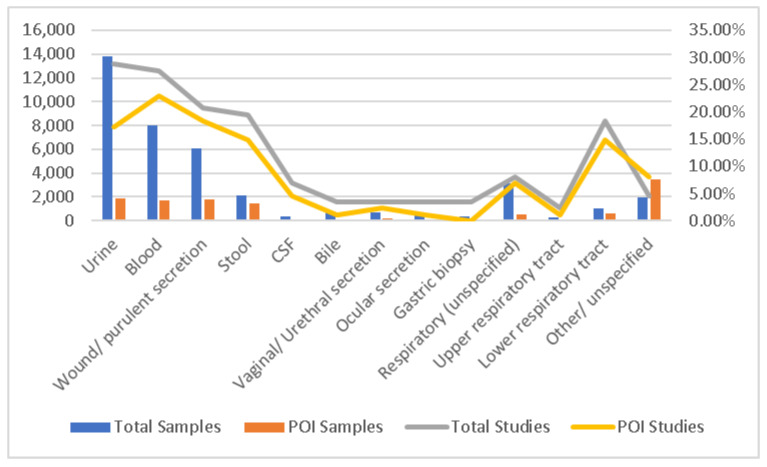
Bacterial isolates in biological samples.

**Figure 4 antibiotics-13-01219-f004:**
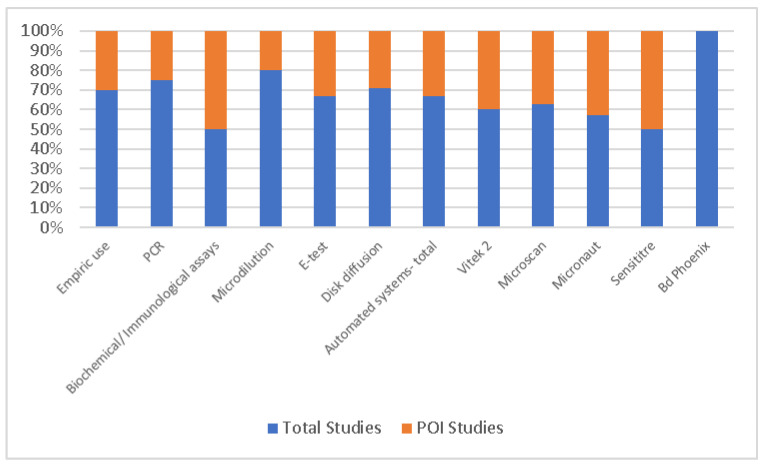
Bacterial isolates in biological samples.

**Table 1 antibiotics-13-01219-t001:** Distribution of bacterial strains across the selected studies.

Pathogen		Total (Individual) n = 87	POI (Individual) n = 49	Total (Cumulated)	POI (Cumulated)
***Staphylococcus*** **spp.**		1223	133 (10.87%)	6847	2632 (38.44%)
	*S. aureus*	4162	1644 (39.50%)		
	CoNS	1462	855 (58.48%)		
***Streptococcus*** **spp.**		700	302 (43.14%)	1105	556 (50.32%)
	Group A/*S. pyogenes*	26	22 (84.62%)		
	Group B/*S. agalactiae*	209	152 (72.73%)		
	Group C	9	8 (88.89%)		
	Group D	17	0 (0.00%)		
	Group G	7	6 (85.71%)		
	Viridans	55	1 (1.82%)		
	*S. pneumoniae*	82	65 (79.27%)		
***Enterococcus*** **spp.**		1449	643 (44.38%)	2416	858 (35.51%)
	*E. faecium*	340	123 (36.18%)		
	*E. faecalis*	625	90 (14.40%)		
	*E. galinarum*	2	2 (100.00%)		
**Enterobacterales**		55	0 (0.00%)	16,951	4447 (26.23%)
	*E. coli*	9653	1988 (20.59%)		
	*Klebsiella* spp.	2664	1226 (46.02%)		
	*K. pneumoniae*	2772	524 (18.90%)		
	*K. oxytoca*	2	1 (50.00%)		
	*Proteus* spp.	951	533 (56.05%)		
**Other Enterobacterales**		3	3 (100.00%)		
	*Enterobacter* spp.	580	108 (18.62%)		
	*Citrobacter* spp.	57	17 (29.82%)		
	*Serratia* spp.	33	19 (57.58%)		
	*Salmonella* spp.	164	27 (16.46%)		
	*Providencia* spp.	15	1 (6.67%)		
	*Morganella* spp.	2	0 (0.00%)		
**NFB**		11	0 (0.00%)	6044	1574 (26.04%)
	*Pseudomonas* spp.	1167	321 (27.51%)		
	*P. aeruginosa*	3365	747 (22.20%)		
	*Acinetobacter* spp.	552	353 (63.95%)		
	*A. baumannii*	930	146 (15.70%)		
	*Stenotrophomonas* spp.	13	7 (53.85%)		
	*Burkholderia* spp.	2	0 (0.00%)		
	*Ralstonia* spp.	4	0 (0.00%)		
**Other GNB**		492	7 (1.42%)	1034	18 (1.74%)
	*Helicobacter* spp.	518	0 (0.00%)		
	*Haemophilus* spp.	24	11 (45.83%)		
**Anaerobes**				3663	1499 (40.92%)
	*Clostridium* spp.	3610	1499 (41.52%)		
	*Bacteroides* spp.	18	0 (0.00%)		
	*Actinomyces* spp.	28	0 (0.00%)		
**Highly pathogenic bacteria**				475	151 (31.79%)
	*Treponema* spp.	18	3 (16.67%)		
	*Mycobacterium* spp.—tuberculosis	457	148 (32.39%)		
**Commensal**		56	2 (3.57%)	65	10 (15.38%)
	*Micrococcus* spp.	1	1 (100.00%)		
	*Moraxella* spp.	8	7 (87.50%)		
**Other**		1180	311 (26.36%)	1223	315 (25.76%)
	*Bacillus* spp.	8	8 (100.00%)		
	*Corynebacterium* spp.	2	0 (0.00%)		
	*Bartonella* spp.	19	0 (0.00%)		
	*Rhizobium* spp.	2	1 (50.00%)		
	*Coxiella* spp.	9	0 (0.00%)		
	*Listeria* spp.	1	1 (100.00%)		
	*Gardnerella* spp.	3	0 (0.00%)		
	*Delftia* spp.	1	1 (100.00%)		
	*Aggregatibacter* spp.	8	1 (12.50%)		

Names in bold reflect totals per group or genus, as obtained from the original articles.

**Table 2 antibiotics-13-01219-t002:** Antibiotic resistance trends.

Category	Antibiotic	Total	Pre-Pandemic	POI	Post-Pandemic
Penicillins	Ampicillin	16 (40.00%)	6 (28.57%)	6 (30.00%)	4 (66.67%)
	Amoxicillin	4 (10.00%)	4 (19.05%)	1 (5.00%)	4 (66.67%)
	Oxacillin	10 (25.00%)	4 (19.05%)	6 (30.00%)	1 (16.67%)
	Piperacillin	7 (17.50%)	1 (4.76%)	4 (20.00%)	3 (50.00%)
	Ticarcillin	2 (5.00%)	1 (4.77%)	1 (5.00%)	0 (0.00%)
	Amoxicillin/Clavulanate	13 (32.50%)	7 (33.33%)	6 (30.00%)	2 (33.33%)
	Piperacillin/Tazobactam	13 (32.50%)	6 (28.57%)	8 (40.00%)	0 (0.00%)
	Ticarcillin/Clavulanate	5 (12.50%)	1 (4.76%)	3 (15.00%)	2 (33.33%)
	Ampicillin/Sulbactam	3 (7.50%)	0 (0.00%)	2 (10.00%)	1 (16.67%)
Cephalosporins	Cefazolin	4 (10.00%)	2 (9.52%)	3 (15.00%)	0 (0.00%)
	Cefoxitin	3 (7.50%)	1 (4.77%)	2 (10.00%)	0 (0.00%)
	Cefuroxime	9 (22.50%)	4 (19.05%)	3 (15.00%)	2 (33.33%)
	Ceftriaxone	9 (22.50%)	5 (23.81%)	5 (25.00%)	1 (16.67%)
	Cefotaxime	8 (20.00%)	4 (19.05%)	4 (20.00%)	1 (16.67%)
	Ceftazidime	19 (47.50%)	10 (47.62%)	13 (65.00%)	2 (33.33%)
	Cefepime	13 (32.50%)	6 (28.57%)	8 (40.00%)	1 (16.67%)
Carbapenems	Meropenem	13 (32.50%)	7 (33.33%)	9 (45.00%)	1 (16.67%)
	Imipenem	14 (35.00%)	7 (33.33%)	11 (55.00%)	2 (33.33%)
	Ertapenem	5 (12.50%)	2 (9.52%)	3 (15.00%)	1 (16.67%)
Monobactams	Aztreonam	6 (15.00%)	3 (14.29%)	3 (15.00%)	2 (33.33%)
Glycopeptides	Vancomycin	8 (20.00%)	3 (14.29%)	4 (20.00%)	2 (33.33%)
	Teicoplanin	6 (15.00%)	3 (14.29%)	5 (25.00%)	0 (0.00%)
Aminoglycosides	Gentamicin	19 (47.50%)	9 (42.86%)	9 (45.00%)	3 (50.00%)
	Amikacin	12 (30.00%)	6 (28.57%)	10 (50.00%)	1 (16.67%)
	Tobramycin	8 (20.00%)	2 (9.52%)	5 (25.00%)	2 (33.33%)
	Streptomycin	3 (7.50%)	1 (4.77%)	2 (10.00%)	0 (0.00%)
Lincosamides	Clindamycin	12 (30.00%)	6 (28.57%)	6 (30.00%)	1 (16.67%)
Macrolides	Erythromycin	14 (35.00%)	7 (33.33%)	6 (30.00%)	2 (33.33%)
	Clarithromycin	2 (5.00%)	1 (4.76%)	1 (5.00%)	1 (16.67%)
Phosphonic acid derivatives	Fosfomycin	7 (17.50%)	6 (28.57%)	5 (25.00%)	1 (16.67%)
Quinolones	Nalidixic acid	1 (2.50%)	1 (4.76%)	1 (5.00%)	0 (0.00%)
	Ciprofloxacin	21 (52.50%)	10 (47.62%)	11 (55.00%)	3 (50.00%)
	Levofloxacin	14 (35.00%)	7 (33.33%)	10 (50.00%)	1 (16.67%)
	Moxifloxacin	4 (10.00%)	1 (4.76%)	2 (10.00%)	1 (16.67%)
	Ofloxacin	6 (15.00%)	2 (9.52%)	4 (20.00%)	0 (0.00%)
Tetracyclines	Tetracycline	13 (32.50%)	6 (28.57%)	5 (25.00%)	2 (33.33%)
	Tigecycline	7 (17.50%)	4 (19.05%)	1 (5.00%)	3 (50.00%)
	Doxycycline	1 (2.50%)	1 (4.76%)	1 (5.00%)	0 (0.00%)
Ansamycins	Rifampicin	6 (15.00%)	3 (14.29%)	3 (15.00%)	1 (16.67%)
Nitrofurans	Nitrofurantoin	6 (15.00%)	5 (23.81%)	4 (20.00%)	1 (16.67%)
Amphenicols	Chloramphenicol	4 (10.00%)	1 (4.77%)	2 (10.00%)	1 (16.67%)
Nitroimidazoles	Metronidazole	2 (5.00%)	1 (4.77%)	0 (0.00%)	1 (16.67%)
Oxazolidinones	Linezolid	8 (20.00%)	4 (19.05%)	5 (25.00%)	1 (16.67%)
Sulfonamides	Cotrimoxazole	17 (42.50%)	8 (38.10%)	10 (50.00%)	1 (16.67%)
Polymyxins	Colistin	11 (27.50%)	5 (23.81%)	7 (35.00%)	2 (33.33%)
	Total studies	40 (45.98%)	21 (24.14%)	20 (22.99%)	6 (6.90%)

**Table 3 antibiotics-13-01219-t003:** List of resistance phenotypes for all 42 studies that mention these data.

Bacteria	MDR	XDR	PDR	Total	Studies (n = 42)
Unspecified	1471 (34.55%)	37 (0.87%)	42 (0.99%)	4258	14 (33.33%)
*Staphylococcus* spp.	1 (12.50%)	0 (0.00%)	0 (0.00%)	8	1 (2.38%)
*S. aureus*	1657 (33.44%)	0 (0.00%)	0 (0.00%)	4955	21 (50.00%)
CoNS	300 (38.96%)	0 (0.00%)	0 (0.00%)	770	5 (11.90%)
*Streptococcus* spp.	56 (11.00%)	0 (0.00%)	0 (0.00%)	509	2 (4.76%)
*Enterococcus* spp.	289 (16.64%)	0 (0.00%)	0 (0.00%)	1737	16 (38.10%)
*E. coli*	2269 (35.59%)	0 (0.00%)	0 (0.00%)	6375	15 (35.71%)
*Klebsiella* spp.	2286 (68.55%)	43 (1.29%)	1 (0.03%)	3335	17 (40.48%)
*Proteus* spp.	170 (33.53%)	1 (0.20%)	10 (1.97%)	507	5 (11.90%)
*Salmonella* spp.	7 (4.70%)	0 (0.00%)	0 (0.00%)	149	2 (4.76%)
*Serratia* spp.	1 (33.33%)	0 (0.00%)	0 (0.00%)	3	1 (2.38%)
*Enterobacter* spp.	18 (16.67%)	0 (0.00%)	0 (0.00%)	108	2 (4.76%)
*Pseudomonas* spp.	683 (46.37%)	9 (0.61%)	1 (0.07%)	1473	13 (30.95%)
*Acinetobacter* spp.	456 (75.12%)	6 (0.99%)	14 (2.31%)	607	9 (21.43%)
*R. picketti*	3 (75.00%)	0 (0.00%)	0 (0.00%)	4	1 (2.38%)
*H. influenzae*	2 (33.33%)	0 (0.00%)	0 (0.00%)	6	1 (2.38%)
*H. pylori*	17 (11.49%)	0 (0.00%)	0 (0.00%)	148	1 (2.38%)
Tuberculosis	2 (2.22%)	0 (0.00%)	0 (0.00%)	90	1 (2.38%)
Other	158 (37.35%)	0 (0.00%)	3 (0.71%)	423	4 (9.52%)

**Table 4 antibiotics-13-01219-t004:** List of resistance phenotypes according to 25 studies explicitly detailing the POI.

Bacteria	MDR	XDR	PDR	Total	Studies (n = 25)
Unspecified	721 (72.98%)	37 (3.74%)	0 (0.00%)	988	7 (28.00%)
*S. aureus*	873 (32.08%)	0 (0.00%)	0 (0.00%)	2721	1 (4.00%)
CoNS	105 (24.88%)	0 (0.00%)	0 (0.00%)	422	3 (12.00%)
*Streptococcus* spp.	2 (1.04%)	0 (0.00%)	0 (0.00%)	192	1 (4.00%)
*Enterococcus* spp.	138 (25.00%)	0 (0.00%)	0 (0.00%)	552	8 (32.00%)
*E. coli*	251 (31.85%)	0 (0.00%)	0 (0.00%)	788	6 (24.00%)
*Klebsiella* spp.	707 (62.29%)	37 (3.26%)	0 (0.00%)	1135	10 (40.00%)
*Proteus* spp.	51 (36.43%)	0 (0.00%)	10 (7.14%)	140	2 (8.00%)
*Serratia* spp.	1 (33.33%)	0 (0.00%)	0 (0.00%)	3	1 (4.00%)
*Pseudomonas* spp.	70 (32.56%)	8 (3.72%)	1 (0.47%)	215	4 (16.00%)
*Acinetobacter* spp.	216 (68.14%)	6 (1.89%)	13 (4.10%)	317	4 (16.00%)
Tuberculosis	2 (2.22%)	0 (0.00%)	0 (0.00%)	90	1 (4.00%)
Other	85 (39.17%)	0 (0.00%)	3 (1.38%)	217	1 (4.00%)

## Data Availability

Data are available upon request from the corresponding author.

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
