# Peer review of "Bacterial Infections, Trends, and Resistance Patterns in the Time of the COVID-19 Pandemic in Romania—A Systematic Review"

_antibiotics, 2024, doi:10.3390/antibiotics13121219_

Round 1
Reviewer 1 Report
Comments and Suggestions for Authors
The authors have undertaken a daunting task of performing an analysis of publications about bacterial infections and resistance in Romania during the pandemic.
As it is now, I feel the article does can not be published.
While interesting for both Romania, the region, and worldwide, the article has multiple issues. The authors seem to have accumulated a vast amount of data from their literary search and are bombarding the reader with all of it. The information does not seem to be systematized, rather it is just thrown at the reader.
The authors should consider what information is relevant and consider which way of conveying the information is best suitable. Apart from information that isn't readable (what is a "bacterial confirmed patient"?), phrases like this: "In total, the wards that were described included infectious diseases and pneumophtysiology with 2342 bacterial confirmed patients and 1672 samples across 19 (21.84%) studies, ICU with 3687 patients and 41 samples across 7 (8.05%) studies, gastroenterology with 1238 patients across 6 (6.90%) studies, general and plastic surgery with 897 patients across 5 (5.75%) studies, cardiology and internal medicine with 223 patients across 4 (4.60%) studies, neurology and dermato-venerology with 41 patients across 4 (4.60%) studies, ophthalmology with 385 patients across 3 (3.45%) studies, pediatrics and N-ICU with 477 patients across 3 (3.45%) studies, obstetrics-gynecology with 1542 patients across 3 (3.45%) studies and others with 2644 patients across 9 (10.34%) studies" should be presented in a figure/table format that is informative.
There are multiple problems reporting strictly the microbiological information. For example, in Table 1, Bacillus and Corynebacterium are listed under Gram Negative Bacteria. Moraxella is listed as a commensal, but CoNS are listed elsewhere. What is the microbiological reasoning for splitting "Enterobacterales" and "Other Enterobacterales"?
What is the reason for having a paragraph re-writing the exact same information from Table 1 in the text? " Cumulatively, for Staphylococcus spp., the grand total was 6847 across all studies, with 2632 (38.44%) being recorded in the manuscripts that properly stratified time periods. There were 4162 strains of S. aureus in total, with 1644 (39.50%) cases occurring during the POI. Coagulase-Negative Staphylococci (CoNS) exhibited a total of 1462 cases, with
855 cases (58.48%) falling within the POI. This results in 1223 individual reported just as." This table takes about a page and a half, and the same information is presented in a narrative format for another page and a half.
Dedicating half a page reporting the "Antibiotic use as therapy" seems useless. The authors seem to count the number of articles which mention a specific type of antibiotic. Should articles reporting solely on C. difficile and M. tuberculosis infections be counted the same way as articles dealing with a wide array of infection types?
Interesting information is presented in an unintelligible way: Information on resistance to antibiotics for all records was declared in 40 (45.98%) articles, with all data being identifiable in the Supplementary Table S5. Among these, the most frequently reported antibiotic categories included penicillins, where ampicillin was mentioned in 37.50% of the articles, followed by amoxicillin (15.00%).
The MOST IMPORTANT methodological flaw of the article, I believe that the authors did not have access to the individual bacteriological records from all the studies cited. They cite a classification for MDR, XDR and PDR, but report MDR, XDR, PDR, CRE, ESBL, MRSA, VRE, etc. Without individual bacteriological records I would be unable to judge an MDR CRO organism from an MDR ESBL producing organism and i believe that including these "extra" phenotypes severely compromises the data. The fact that Table S5 lacks other data, which need to be taken from previous tables, makes it useless, due to the fact that you cannot interpret the type of infection that they are addressing and the number of cases.
From a data reporting point of view, Table 3 is wasteful because combining Gram positive and negative bacteria together forces the author to report, for example, 0% MRSA phenotype for E. coli. Moreover, how can there be an "Overall" category with 7 studies, while the top of the table reports a total of 25 studies? How can S. aureus be at the same time MRSA and MRCoNS? And why report only 2 as MDR, when it is obvious for all that hospital acquired MRSA strains are most often resistant to other antibiotic classes.
I have tried parsing Table S5 and have found very different ways of reporting data on susceptibility (Klebsiella spp.: ampi/ sul -3/ 15 for study 72, S. aures: Oxacillin -3 for study 74 and even cefotaxime -99, 137% for article 81). This does not look like sound data gathering and I do not believe that the reporting that comes from such data can be reasonably analyzed.
As for the section on Discussions, this should offer a broader discussion, reflecting on the data and citing adequate resorces that are relevant, not only one study/report per statement.
Some statements are seem to contradict themselves if not explicitly discussed: "Carbapenems, led by meropenem (27.50%), were frequently used to treat multidrug-resistant infections, particularly CRE." - Carbapenems were frequently used to treat Carbapenem Resistant Enterobacteria.
While some of the issues pointed out are addressed in the Limitations section, I don't believe that this should be a way for the authors to excuse bad methodology.
Comments on the Quality of English LanguageThere are a number of issues regarding how this article is written:
- for example, period of interest (later referred to as POI) is never explicitly abbreviated in this fashion in the article, but then the acronym is used extensively, and I believe this makes following the information difficult.
- Similarly, the terms CRE and CRO seem never to have been specifically abbreviated.
- phrases like "nature of the articles" are to be avoided and a better suited "article type" should be used instead.
- EXTENSIVE rewriting is needed to properly convey information from long and difficult, seemingly convoluted phrases like: "Based on the 70 only patients’ studies, there were a total of 21527 patients, from which 14288 (66.37%) had a confirmed infection, while on account of the 10 only samples/ episodes studies there were 18419, while the 7 studies that described both metrics pooled together 3941 bacterial confirmed patients and 6371 samples/ episodes."
Author Response
Thank you for your suggestions. Please find the responses attached.

Reviewer 2 Report
Comments and Suggestions for Authors
Thanks for giving me a chance to review. Very interesting to see this study from Romania. It is a very comprehensive study about the landscape of bacterial infections in Romania. Here are some suggestions to make the study more interesting:
- Line 74: Are men a vulnerable population? If yes, please provide a reference.
- Please spell out POI. What does it indicate?
- Compared to other countries with similar socioeconomic standards and healthcare systems where do the major differences lie in Romania?
- Could there be a specific sub-focus on infections in elderly individuals and pregnant women?
Author Response
Dear reviewer 2, thank you for your time. we greatly appreciate you help. Here are the responses for each comment:
-
- Line 74: Are men a vulnerable population? If yes, please provide a reference.
- The reference is at the end of the paragraph, and is specifically about the vulnerability of romanian patients towards COVID-19, no. [3] (Mitrică, B.; Mocanu, I.; Grigorescu, I.; Dumitraşcu, M.; Pistol, A.; Damian, N.; Şerban, P. Population Vulnerability to the SARS-CoV-2 Virus Infection. A County-Level Geographical-Methodological Approach in Romania. GeoHealth 2021, 5, e2021GH000461.
- Please spell out POI. What does it indicate?
- POI is the Period of interest, reffering to the years 2020-2022, as mentioned in the materials section. Indeed, we overlooked adding the parentheses, which have now been added.
- Compared to other countries with similar socioeconomic standards and healthcare systems where do the major differences lie in Romania?
- Thank you for the suggestion. Such information has been added throughout the discussion section for countries like Serbia, Poland, Hungary, Czech Republic and even Western European areas like Greece, Italy or the Netherlands.
- Could there be a specific sub-focus on infections in elderly individuals and pregnant women?
- Thank you for the suggestion. As other reviewers have pointed that the text requires some content adjustment, we also feel this would overload our manuscript. However, these ideas, alongside the study of the infections during the pandemic in children are part of the doctoral studies of the first author, which will be tackled in subsequent papers.
Reviewer 3 Report
Comments and Suggestions for Authors
This study has included a comprehensive set of 87 studies, providing abundant evidence on the bacterial and antibiotic resistance trends in Romania during the COVID-19 pandemic. However, the Introduction section could be improved by providing more relevant background information.
The current introduction includes some non-relevant details, such as information about COVID-19 vaccine hesitancy, which does not directly relate to the focus of the study. The introduction should be reorganized to better establish the importance and necessity of this systematic review.
It would be helpful if the authors provided some baseline information on the antimicrobial resistance (AMR) situation in Romania prior to the pandemic. This would allow readers to better contextualize the severity of the AMR trends observed during the COVID-19 period. As it stands, the study can only demonstrate the high levels of AMR during the pandemic, without a clear comparison to the pre-pandemic conditions.
Author Response
Dear Reviewer 3, thank you for taking your time and assessing our manuscript. Here are the responses for your comments:
- The current introduction includes some non-relevant details, such as information about COVID-19 vaccine hesitancy, which does not directly relate to the focus of the study. The introduction should be reorganized to better establish the importance and necessity of this systematic review.
- Thank you, the introduction has been reworked and non-relevant data were cut.
- It would be helpful if the authors provided some baseline information on the antimicrobial resistance (AMR) situation in Romania prior to the pandemic. This would allow readers to better contextualize the severity of the AMR trends observed during the COVID-19 period. As it stands, the study can only demonstrate the high levels of AMR during the pandemic, without a clear comparison to the pre-pandemic conditions.
- Thank you. This information was added and contextualized both in the introduction and in the discussion sections.